# Comprehensive Analysis of BR Receptor Expression under Hormone Treatment in the Rubber Tree (*Hevea brasiliensis* Muell. Arg.)

**DOI:** 10.3390/plants12061280

**Published:** 2023-03-11

**Authors:** Bingbing Guo, Longjun Dai, Hong Yang, Xizhu Zhao, Mingyang Liu, Lifeng Wang

**Affiliations:** Rubber Research Insititute, Chinese Academy of Tropical Agriculture Sciences, Haikou 571101, China

**Keywords:** brassinosteroids, BRI1, BAK1, receptor-like kinase, rubber tree

## Abstract

Brassinosteroids (BRs) are important for plant growth and development, with BRI1 and BAK1 kinases playing an important role in BR signal transduction. Latex from rubber trees is crucial for industry, medicine and defense use. Therefore, it is beneficial to characterize and analyze *HbBRI1* and *HbBAK1* genes to improve the quality of the resources obtained from *Hevea brasiliensis* (rubber tree). Based on bioinformatics predictions and rubber tree database, five *HbBRI1s* with four *HbBAK1s* were identified and named *HbBRI1*~*HbBRL3* and *HbBAK1a*~*HbBAK1d*, respectively, which were clustered in two groups. HbBRI1 genes, except for *HbBRL3*, exclusively contain introns, which is convenient for responding to external factors, whereas *HbBAK1b/c/d* contain 10 introns and 11 exons, and *HbBAK1a* contains eight introns. Multiple sequence analysis showed that HbBRI1s include typical domains of the BRI1 kinase, indicating that HbBRI1s belong to BRI1. HbBAK1s that possess LRR and STK_BAK1_like domains illustrate that HbBAK1s belong to the BAK1 kinase. BRI1 and BAK1 play an important role in regulating plant hormone signal transduction. Analysis of the cis-element of all HbBRI1 and HbBAK1 genes identified hormone response, light regulation and abiotic stress elements in the promoters of *HbBRI1s* and *HbBAK1s*. The results of tissue expression patterns indicate that *HbBRL1/2/3/4* and *HbBAK1a/b/c* are highly expressed in the flower, especially *HbBRL2-1*. The expression of *HbBRL3* is extremely high in the stem, and the expression of *HbBAK1d* is extremely high in the root. Expression profiles with different hormones show that HbBRI1 and HbBAK1 genes are extremely induced by different hormone stimulates. These results provide theoretical foundations for further research on the functions of BR receptors, especially in response to hormone signals in the rubber tree.

## 1. Introduction

Brassinosteroids (BRs) belong to a group of plant steroid hormones and were first found in the pollen of Brassica napus in the 1970s [1]. They not only play an important role in cell elongation, differentiation, flowering, pollen development and seed size, but also play a more prominent role in the process of plant resistance [2,3,4]. Receptor-like kinases (RLKs) located in the plant cell surface transmit regulation signals [5]. BRI1 is the receptor of BR, which is a leucine-rich repeat receptor-like kinase (LRR-RLK) that is combined with the extracellular ligand-binding domain, transmembrane domain and intracellular kinase domain [6].

BR-insensitive 1 (BRI1) and BRI1-associated kinase receptor 1 (BAK1) are located in the plasma membrane, where they transmit the BR signal to the cytoplasm through a series of phosphorylation and dephosphorylation reactions, which are then transferred to the nucleus to ultimately promote/restrain the gene expression of target genes. Research has shown that BRI1 and BAK1 are important kinases for perceiving and transferring the BR signal to different domains that possess different functions. For BRI1, the extracellular domain constitutes of 25 LRRs, and the BR recognition domain is an island domain formed by 70 amino acids located between the 21st and 22nd LRRs; meanwhile, the intracellular kinase domain is divided into 12 conserved domains, whereas the transmembrane domain connects the extracellular and intracellular domains together [2,7,8,9]. BAK1 contains the N-terminal signal peptide, five LRRs, the transmembrane domain and the cytoplasmic serine/threonine kinase domain [10]. Additionally, BR-insensitive-like 1 (BRL1) and BR-insensitive-like 3 (BRL3) are the homologous proteins of BRI1 involved in BR signal transduction [11,12].

The mutation of BRI1 in *Arabidopsis* indicated that the function of the island domain and kinase domain of BRI1 was very important in signal transduction [13,14]. When the island domain mutates, it can lead BRI1 to decrease or even lose the perception of the BR signal [8,9,15]. Overexpressed *BAK1* was found to partially restore the BRI1 mutant phenotype [16]. The activity of the BRI1 and BAK1 kinases plays a crucial role in the transduction of the BR signal from the cell membrane to the nucleus. During BR signaling, BRI1 combines with BAK1 to form a receptor kinase complex, and phosphorylation occurs reciprocally to trigger subsequent reactions. The association of BRI1 and BAK1 is mediated by both kinases and is dependent on BRI1 kinase activity [4]. If mutations occur in the structural domain of the kinase, they prevent the downward transmission of BR signals that exhibit BR insensitivity. BRI1 and BAK1 transfer the signal to BES1/BZR1 to bind promoters of downstream genes and regulate the expression of target genes of BR, and thus regulate various growth and development progressions [17]. On the other hand, BES1 and BZR1 are combined with promoters of biosynthesis genes such as CPD (constitutive photomorphogenesis and dwarfism) in the nucleus, to restrain the synthesis of BR, achieving dynamic regulation at the hormone level [18]. In the absence of BRs or low levels, the C-terminals of BRI1 and BSK1 (BR-signaling kinase) bind to BRI1 to inhibit the interaction between BRI1 and BAK1, and BIN (BR-insensitive) is retained in the cytoplasm by 14-3-3 proteins, thereby inhibiting BR signal transduction and perception [19]. Therefore, the quantity of BRs and their receptor proteins is regulated by a strict dynamic equilibrium mechanism.

Up to now, the BRI1 and BAK1 receptor kinases have been identified and their mechanism in other species has been illustrated [16,20,21,22]. The rubber tree is originally a plant from the tropical jungle of the Amazon basin in South America, where the locals call it “Hevea”. It is the world’s main source of natural rubber. However, the function of BRs in rubber trees is complex and poorly understood. Hence, authenticating the BR signaling pathway transduction genes in the rubber tree genome, and speculating on their potential functions will provide useful information. Interspecific differences might lead to the evolution and functional diversification of *HbBRI1* and *HbBAK1* members. In this study, *HbBRI1* and *HbBAK1* members were identified based on a rubber tree genome database, which allowed for characterizing their physical and chemical properties, gene structure, cis-acting regulatory elements and expression pattern in different organs. In addition, *HbBRI1* and *HbBAK1* responses to different hormones at different times were preliminarily studied and used to analyze their diverse functions. Hence, our findings provide a basis for the molecular and functional study of BRI1 and BAK1 genes in rubber tree.

## 2. Results

### 2.1. Identification of BRI1 and BAK1 Genes in Rubber Tree

In this study, a total of five and four complete overviews of HbBRI1 and HbBAK1 genes, respectively, were obtained from a rubber tree using all kinds of bioinformatics resources as candidates with homologies to *Arabidopsis thaliana* and *Oryza sativa* [11,23] (Table 1). The HbBRI1 and HbBAK1 genes were named *HbBRI~HbBRL3* and *HbBAK1a/b/c/d*, respectively, according to their locus numbers. In these genes, only *HbBRI1* and *HbBAK1a* were located in the forward strand, and the others were located in reverse strands. The full length of the open reading frames (ORFs) was from 1206 to 3688 bp, and the amino acids ranged from 402 to 1228 aa. The predicted PI values ranged from 5.41 to 8.15. Based on the GRAVY values, we could determine that the HbBRI1s and HbBAK1s were hydrophilic proteins, except for HbBAK1a.

### 2.2. Phylogenetic Analysis of BRI1 and BAK1 Members

A total of 47 amino acid sequences were used to create an NL phylogenetic tree to determine the phylogenic relationship of BRI1 and BAK1 with rubber trees (9) *A. thaliana* (9), *Oryza* (6), *Populus* (9), *Citrus* (6), *Thebroma* (4) and *Durmurr* (4) (Figure 1). According to the bootstrap values and topology of the phylogenetic tree, BRI1s and BAK1s from seven species were divided into two groups: group I and group II. Group II was more ancient than group I. Group I contained HbBRI1s which could be separated into three subgroups, and *HbBRI1*, which is the homologous gene of *AtBRI1*. BAK1 was divided into two subgroups, where *HbBAK1c/d* were the homologous genes of *AtBAK1*.

### 2.3. Sequence Characteristics of HbBRI1 and HbBAK1 Members

The amino acid sequences of HbBRI1s and HbBAK1s contain some characteristic domains to conduct kinase functions (Figure 2 and Figure 3). Four main domains were found in HbBRI1 and HbBAK1 protein sequences: the signal peptide, LRR domain, transmembrane domain and cytoplasmic kinase domain. Regarding gene structure, only *HbBRL3* contained two introns, whereas the others had no introns. However, there were 10 introns in *HbBAK1b/c/d*, and eight introns in *HbBAK1a* alone (Figure 4A). Additionally, the conserved motifs in the HbBRI1s and HbBAK1s shared a consistent distribution pattern (Figure 4B). In contrast to HbBRI1, the amino acid sequence of HbBAK1a was missing motif 2/4/7/9. Conserved motifs are shown in Figure 4C. All these results show that the BRI1 and BAK1 genes in the rubber tree are very conservative, but during evolution, HbBAK1a had a missing domain in the C-terminal.

### 2.4. Structural Domain and Cis-Elements in HbBRI1 and HbBAK1

Based on the HMMER software, we determined that all HbBRI1 and HbBAK1 proteins possessed an LRR domain and a Pkinase domain with active sites, except for HbBAK1a, which had no active sites (Figure 5A). The amino acid sequences all started with the LRRNT-2 domain and ended with the Pkinase domain, indicating that HbBRI1 and HbBAK1 are plant leucine-rich repeat receptor-like protein kinases and play an important role in plant growth and hormone signal transduction. In order to understand the cis-elements in HbBRI1 and HbBAK1, we predicted the cis-elements based on the promoter sequences of *HbBRI1* and *HbBAK1*, using PLACE; the results are shown in Figure 5B. These five *HbBRI1* and four *HbBAK1* prompters included the conserved core element TATA-box and enhanced the element CAAT-box, which has similar basic structural characteristics as the eukaryotic gene promoter and contains many elements that respond to light, illustrating that light may have a significant effect on HbBRI1 and HbBAK1 genes, such as on the 3-AF1 binding site, GT1, G-box, MRE, 4cl-CMA2b, ATCT motif, Gap-box and AT-rich. Additionally, HbBRI1 and HbBAK1 contained some elements that respond to hormones: GARE, P-box, TATC-box and CGTA-box responds to GA; the ABRE and AAGAA motif responds to ABA; the SARE and TCA element responds to SA; and the TGACG motif and CGTCA motif responds to JA. These responses indicate that BRI1 and BAK1 are not only regulated by BR, but also by other hormones. At the same time, HbBRI1 could respond to some abiotic stress through DRE-core, LTR, MBS, MYB, MYC, GC-motif, WUN-motif and TC-rich repeat cis-elements. The O_2_ site is involved in secondary metabolism, ARE is involved in the antioxidative response to maintain cell homeostasis and GCN4 motif is essential for endosperm-specific gene expression in HbBAK1s. Thus, HbBRI1 and HbBAK1 are important links between light signal transduction and hormone signal transduction.

### 2.5. Expression Analysis of HbBRI1 and HbBAK1 Genes in Different Tissues

Combined with the RPKM value from the rubber tree in NCBI, with the accession code of SRP069104, we analyzed the expression patterns of *HbBRI1s* and *HbBAK1s* in different tissues of CATAS73397 (Appendix A). HbBRI1 and HbBAK1 genes displayed spatial specificity in different tissues. *HbBRL2-1* and *HbBRL2-2* were expressed higher in the stem; *HbBRL3* had a high expression level in the seed; *HbBRL1* and *HbBAK1d* were highly expressed in the secondary laticifer, which is a tissue that synthesizes latex; and *HbBRI1* and *HbBAK1c* were highly expressed in the primary laticifer that appeared in the seedling. Based on these findings, it can be concluded that HbBRI1s and HbBAK1s play an important role in latex synthesis. By using real-time fluorescence quantification, different tissues in the rubber tree were employed (root, stem, leaf, flower, branch, latex), and we found that nearly all genes had the highest expression in the flower, with *HbBRL3* and *HbBAK1d* being the exceptions, and nearly all genes had the lowest expression in latex, with the exception being *HbBRL3* and *HbBRL2-2* (Figure 6). Notably, for *HbBRL2-1*, the relative expression level reached 575-fold higher than the control. The expression levels of *HbBRL2-2/5* and *HbBAK1c* were more than 300-fold higher than the control.

### 2.6. Expression Profile of HbBRI1 and HbBAK1 Response to Hormone Stress

BRI1 coordinates the crosstalk between BRs and other hormones. qRT-PCR was used to analyze the expression profile of *HbBRI1s* and *HbBAK1s* that was induced by different hormones, e.g., abscisic acid (ABA), brassinolide (BR), ethephon (ETH), gibberellin (GA), jasmonic acid (JA) and salicylic acid (SA) (Figure 7). Based on the result, we determined that the HbBRI1 and HbBAK1 genes show varying expression patterns in response to different hormone treatments. When the rubber tree is treated with ABA, *HbBRL1*, *HbBRL2-1/2* and *HbBAK1a/b* had an extraordinarily high expression after 10 h. This was especially true for *HbBAK1b*, as its expression profile was 269 times more than the control. The peak time of *HbBRI1* and *HbBAK1c/d* was 24 h, and 6 h for *HbBRL3*. Under BR treatment, the expression level of HbBRI1s over time exhibited specific decreasing and increasing. All HbBRI1 genes showed a high expression level after 10 h, except *HbBRL1*. *HbBRL2-2* was strongly induced by the BR treatment after 10 h, with an expression level more than 81 times that of the control. As opposed to *HbBRI1*, the expression level of *HbBAK1c/d* first increased, then decreased and, finally, increased again; whereas, *HbBAK1a/b* first decreased, then increased and, finally, decreased again under different BR treatment methods. With ETH treatment, all HbBRI1s were upregulated at different time points and reached their peak within 10 h: *HbBAK1a/b* were upregulated in 10 h, and *HbBAK1c/d* were upregulated in 24 h. The expression level of the HbBRI1s was significantly downregulated by GA and JA treatments at all time points, but upregulated for all *HbBAK1s* by the JA treatment, except for *HbBAK1a*. In response to the SA treatment, the expression levels of the five HbBRI1s were all upregulated and demonstrated variation tendencies over time. *HbBRI1* was extremely induced by SA, and its peak value was more than 290 times that of the control. *HbBAK1a/b* exhibited a peak value within 10 h and *HbBAK1c* within 2 h when processed with SA, but *HbBAK1d* was downregulated.

## 3. Discussion

In recent years, there has been more research on BRs. The functions of BRs include cell enlargement and division, senescence manipulation, male fertility, pollen development and fruit ripening, covering almost all of the plant growth and development processes [3,24,25]. Therefore, the important role of BRs in plant hormones has been established. BRI1 and BAK1 have been widely studied as BR receptors that perceive and transmit BR signaling. Identifying and exploring the physiological and biochemical characteristics of BRI1 and BAK1 in the rubber tree, by applying a theoretical foundation, can be utilized to breed high-quality varieties. In the current study, we conducted genome-wide research on BRI1 and BAK1 genes in the rubber tree, and also performed a comprehensive analysis of the HbBRI1 and HbBAK1 gene responses to hormone stresses. Five HbBRI1 and four HbBAK1 genes were identified in the rubber tree, and we analyzed their physicochemical properties, phylogenetic characteristics, conserved domain and motif, gene structure, cis-elements and response to hormones. Four BRI1- and five BAK1-associated genes in *Arabidopsis thaliana* [7,26], as well as five BRI1- and two BAK1-associated genes in *Oryza sativa* [23,27], were compared to the five BRI1 and four BAK1 genes in the rubber tree. The number of BRI1 and BAK1 genes in different species showed no significant difference, indicating that BRI1 and BAK1 are evolutionarily conserved and ancient in plant growth and development. Additionally, the PI value of all HbBRI1s and HbBAK1s was no more than seven, except in HbBAK1a, indicating that HbBRI1s with HbBAK1s are more fit for a saline environment. Furthermore, all HbBRI1 and HbBAK1 proteins possessed a transmembrane domain that was located in the plasma membrane, illustrating that BRI1 and BAK1 deliver the BR signal to the plasma membrane.

Since orthologues have similar biological characteristics, phylogenetic analysis was used to predict their function in the evolutionary process. Twenty-five BRI1s were divided into three subgroups, and twenty-two BAK1s were divided into two subgroups. The results of this study are consistent with those of previous studies. AtBRI1 and OsBRI1 encouraged precocious flowering, delay senescence and promoted the elongation of leaves and petiole, while enhancing the sensitivity to BRs [9,14,28,29]. HbBRI1 was clustered with AtBRI1, and thus may share the same functions as subgroup Ib. BRL1 and BRL3 are receptors which are unnecessary, but can be combined with BRs. AtBRL1, AtBRL3, OsBRL1 and OsBRL3 are involved in vascular differentiation, cell elongation and cell division [11,23]. HbBRL1 and HbBRL3 were clustered with AtBRL1 and AtBRL3, and thus may show similar functions as subgroup Ia. HbBRL2-1, HbBRL2-2, AtBRL2 and OsBRL2 belonged to the group Ic. BRL2 was unnecessary and could not be combined with the BR that interacted with vascular-specific adaptor proteins to influence leaf venation [30]. These results provide a foundation for the further study of the functions of BRI1 genes in the rubber tree. The overexpression of *BAK1* resulted in elongated organ phenotypes, whereas a null allele of *BAK1* displayed a semi-dwarfed phenotype and had reduced sensitivity to BRs [31]. AtBAK1 worked with AtBRI1 to transfer the BR signal, but functioned independently to the BR signal to regulate immune response and apoptosis. The synergistic effect of the BR signal and immune response required BAK1, indicating that there is a complex crossover between the BR signal and immune response that involves BAK1 [32]. *OsBAK1* is expressed in seedling leaves, stems and sword leaves, which can be activated by blast fungus, host cell death, defense signaling molecules (such as SA and JA) and other stress signals. The overexpressed *OsBAK1* gene in rice can increase blast resistance [33]. HbBAK1 is the homologous gene of AtBAK1 and OsBAK1, and may share the same or similar functions.

Moreover, we aligned the multiple BRI1 and BAK1 amino acid sequences from the rubber tree, revealing four domain structures, such as the signal peptide, LRR domain, transmembrane domain and cytoplasmic kinase domain. These four domains are conserved and important in BRI1 and BAK1 genes [34,35,36,37,38]. Based on the results of gene structure, we determined that *HbBRL1*, *HbBRI1*, *HbBRL2-1* and *HbBRL2-2* have no introns and showed a similar gene structure, indicating that these genes were ancient, conserved and had no reason to develop intron shearing that could rapidly respond to external factors.

More evidence has been found indicating that genes which contain response elements are closely related to environmental changes [39]. BRI1 and BAK1 are the receptors of the BR signal pathway that are involved in multiple stress responses. From the results of our cis-element prediction, we determined that *HbBRI1* and *HbBAK1* members contained many elements to respond to hormone and abiotic stress, including ABA, GA, SA, JA, light, wounds, drought, etc. All these results indicate that BRI1 and BAK1 are LRR-RLKs that regulate hormone signaling transduction and abiotic stresses. BR has various effects in plants when it interacts with other hormones [40,41]. BR and other hormones (ABA, ETH, GA, JA and SA) promote and antagonize each other under disparate conditions [42,43,44,45,46,47,48]. BRI1 and BAK1 can be qualified as upstream components of the BR signal [49,50,51]. The mutant of *BRI1* is insensitive to BR, but treatment with other hormones has no significant differences [52]. The mutant of *BRI1* in tomato is constantly sensitive to GA and ABA [53]. However, *uzu*, a mutant of *BRI1* in barley, was more sensitive to ABA in the seed germination rate. In *A. thaliana*, *bri1-301* is a weak mutant plant of *BRI1* and is more sensitive to ABA [52]. Overexpressed *BAK1* could complement the phenotype of *bri1-5*, which had an amino acid mutant in the extracellular domain, but could not complement the phenotype of *bri1-4*, which functionally incapacitated the kinase domain, indicating that BAK1 and BRI1 exist as functional complements, but require the kinase activity of BRI1 [37]. As a BR receptor, BAK1 is involved in plant innate immune response, abiotic stress and cell death. BAK1 interacts with AvrPto and AvrPtoB to initiate effector-triggered immunity [54]. In apples, *MdBAK1* regulates sugar, hormones, BR with ethylene, lignin and the adversity response to promote the growth of apple saplings [55]. In this study, we treated different hormones found in the rubber tree to detect the regulating patterns and filtrate the response factors of *HbBRI1s* and *HbBAK1s*. In apples, *MdBRI1/2/4* and *MdBAK1* are extremely induced by BR [55,56], which was similar to our research. In this work, the expression level of HbBRI1s was upregulated by ABA, BR and ETH, and was downregulated by GA and JA. Combined with microarray analysis and qRT-PCR results in the rubber tree, we found that the expression of *HbBRI1s* and *HbBAK1s* in different tissues and various treatments shared differential functions, which were similar with those found in other species [52,53,57,58]. We assumed that the strong induction regulation mechanism appeared after 10 h with the ABA, BR and ETH treatment, after 6 h with the SA treatment, after 2 h with the GA treatment, and after 0.5 h with the JA treatment. In addition, *HbBRL1*, *HbBRL2-2*, *HbBRL2-1* and *HbBRI1* were most strongly induced by ABA, BR, ETH and SA, respectively. HbBAK1 genes were all up/downregulated by hormones to varying degrees, and these four HbBAK1 genes showed different patterns at different times. Notably, the *HbBAK1b* response to ABA reached a 269-fold increase in 10 h. For the GA treatment, the expression levels of *HbBAK1s* were all upregulated in 0.5 h, then fell, and finally rose after 10 h. Unlike HbBRI1, the expression patterns of all HbBAK1 gene responses to hormones were irregular under the GA treatment. *HbBAK1a* was downregulated and *HbBAK1b/c/d* were upregulated by JA. Besides *HbBAK1d*, the HbBAK1 genes were induced by SA. Additionally, HbBRI1s were more strongly induced by BR than HbBAK1s, especially *HbBRI14*. We speculated that HbBRI1 with HbBAK1 had a feedback-regulating effect on BR. Ethylene can promote latex yield [59]. BR increased the expression of BRI1 and BAK1 to promote synthesis of ETH [60]. Under ETH and BR treatment, the expression level of all *HbBRI1s* and *HbBAK1s* was increased. Above all, we deduced that HbBRI1 and HbBAK1 were involved in latex synthesis. Hence, we speculate that HbBRI1 and HbBAK1 genes have a functional diversity and are regulated by various hormones in the plant growth and development process, most likely sharing a similar expression pattern within the same subgroup, because they are more closely related.

## 4. Materials and Methods

### 4.1. Retrieving BRI1 and BAK1 Genes from the Rubber Tree

Amino acid sequences of BRI1 and BAK1 members in *Arabidopsis thaliana* and *Oryza sativa* were extracted from the TAIR database (15 September 2021) (https://www.arabidopsis.org/index.jsp) and TIGR database (15 September 2021) (http://rice.plantbiology.msu.edu) as queries to blast in the rubber tree database (20 September 2021) (http://hevea.catas.cn/tool/v1/toBlast) and the NCBI database (21 September 2021) (http://www.ncbi.nlm.nih.gov/), using default parameters. If the sequence satisfied E<10-10, it was taken as a candidate HbBRI1 and HbBAK1 protein. Pfam (21 September 2021) (http://pfam.xfam.org/search/sequence) and SMART (21 September 2021) (http://smart.embl.de/) online databases were used to affirm the existence of conserved domains. Sequences which lacked the typical domains were rejected. A total of 5 BRI1 and 4 BAK1 members were identified. Molecular weight, isoelectric points and GRAVY values were estimated by ExPASy (23 September 2021) (http://web.expasy.org).

### 4.2. Sequence Alignment and Phylogenetic Analysis of the HbBRI1 and HbBAK1 Genes

Candidate sequences were aligned via Clustal W (10 February 2022) (https://www.genome.jp/tools-bin/clustalw) and MUSCLE (10 February 2022) (https://www.ebi.ac.uk/Tools/msa/muscle/), to perform multiple sequence alignment. The phylogenetic tree was reconstructed by using the Molecular Evolutionary Genetics Analysis version (MEGA) X software, and employing the neighbor-joining (NJ) method with the Poisson model and 1000 replicates, where the “Branch Swap Filter” was set to “Very Strong”, the “Rates among sites” parameter was set to uniform and the “Number of Threads” was 1.

### 4.3. Gene Structure and Protein Motif Analysis of BRI1 Family in the Rubber Tree

We used the Gene Structure Display Server 2.0 (GSDS, http://gsds.cbi.pku.edu.cn) to identify gene structures by CDS sequences and genomic sequences. MEME v5.5.1 (https://meme-suite.org/meme/tools/meme, accessed on 7 March 2023) was used to predict conserved motifs in HbBRI1s and HbBAK1s, where the motif number was set to 10.

### 4.4. Conserved Domain and Cis-Element Analysis

The prediction of conserved domains in HbBRI1 and HbBAK1 proteins was performed with the HMMER v3.3.2 online software (http://www.hmmer.org/). Based on rubber tree genome data, we extracted sequences with a length of 2000 bp upstream of all HbBRI1 and HbBAK1 genes as promoter sequences, and used the PLACE (https://www.dna.affrc.go.jp/PLACE/?action=newplace, accessed on 7 March 2023) online software to analyze promoter sequence characteristics and cis-elements.

### 4.5. Plant Materials, Treatment and qRT-PCR Assay

The rubber tree cultivar, CATAS73397, was planted on the experimental farm at of the Chinese Academy of Tropical Agricultural Sciences, Danzhou City, Hainan Province, China (19°51′51 N; 109°55′63 E). Samples from different tissues, including root, stem, leaves, branch, latex and flowers, were collected from 15-year-old healthy tapping trees and used for tissue-specific expression analysis. Tissue culture seedlings of CATAS73397 were grown in the Hainan Natural Rubber New Planting Material Innovation Base, and treated with the following hormones for our preliminary experiment [61]: 100 μmol/L SA (salicylic acid), 200 μmol/L JA (methyl jasmonate), 200 μmol/L ABA (abscisic acid), 1% (*v*/*v*) ETH (ethephon), 200 μmol/L GA3 (gibberellin) and 5 μmol/L BR (brassinolide). Each sample included three independent biological replicates. After hormone treatment, leaf samples were collected at 0 h, 0.5 h, 2 h, 6 h, 10 h and 24 h for testing of gene relative expression levels. Total RNA was extracted by the RNAprep Pure Plant Plus Kit (Tiangen Biotech, Beijing, China), and 1 μg of all samples was reverse-transcribed to cDNA with the RevertAid Reverse Transcriptase (Thermo Scientific, Waltham, MA, USA), which was taken for qRT-PCR via CFX96 (Bio-Rad, Hercules, California, USA). qRT-PCR was carried out by the following steps: preparation of reaction solution (8.2 µL of ddH_2_O, 0.4 µL od forward primer, 0.4 µL of reverse primer, 10.4 µL of cDNA and 2× 10 µL of ChamQ Universal SYBR qPCR Master Mix (Vazyme, Nanjing, China), followed by real-time PCR reaction (stage 1: 95 °C for 30 s; stage 2: 95 °C for 10 s and 60 °C for 30 s, up to 40 repetitions; and stage 3: 95 °C for 15 s, 60 °C for 60 s and 95 °C for 15 s). Three biological replicates were performed for each sample. Expression levels were calculated by 2^−△△Ct^.

## 5. Conclusions

In conclusion, a total of five BRI1 and four BAK1 genes were identified in the rubber tree, and we analyzed their genetic information, such as gene structure, cis-elements, conserved domains, etc. Additionally, the spatio expression differences in this family and the expression patterns under hormone stresses were studied. Finally, it was found that the expression levels of *HbBRL1*, *HbBRI1*, *HbBRL3* and *HbBAK1b* were relatively high in different tissues and in response to different hormones. Due to the limited research on BRI1 and HbBAK1 members in the rubber tree, these genes will become important candidate genes for future research on the mechanism of hormone response, and for developing new varieties of the rubber trees.

## 6. Patents

There are no patents resulting from the work reported in this manuscript.

## Figures and Tables

**Figure 1 plants-12-01280-f001:**
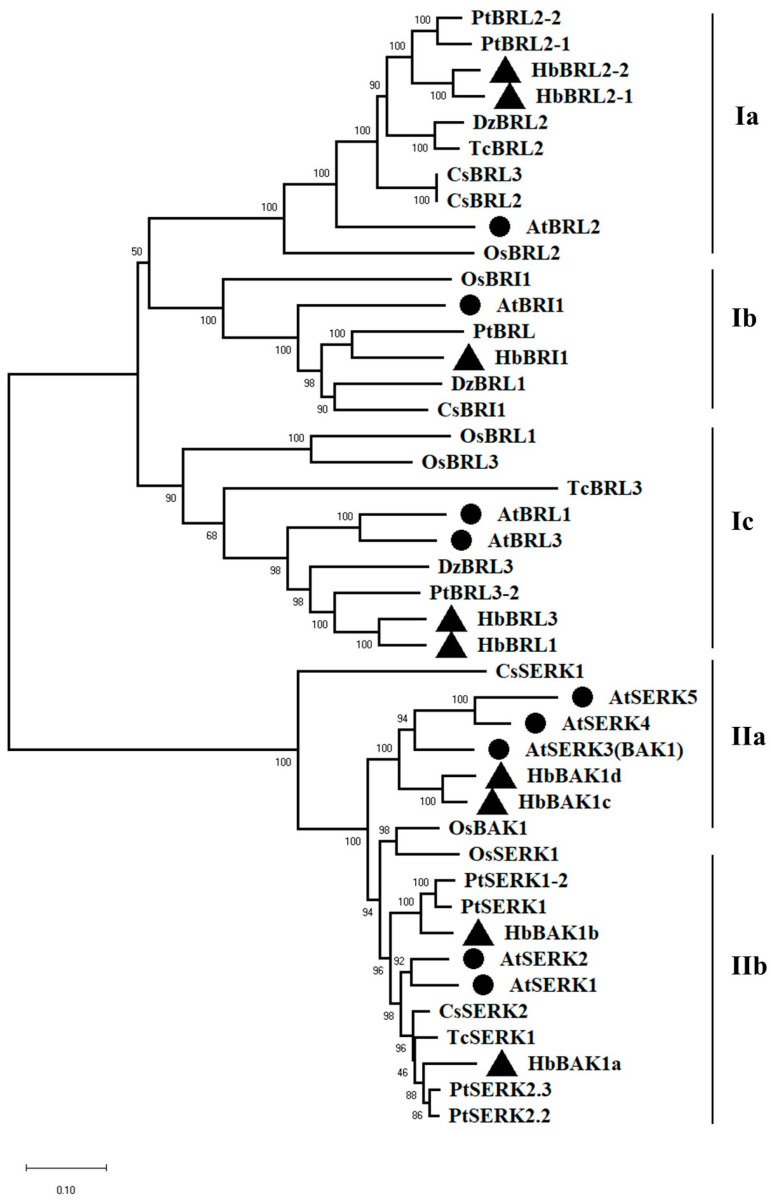
The phylogenetic tree of BRI1 and BAK1 proteins in *Arabidopsis*, rice, durian, aspen, cacao, orange and rubber tree. Phylogenetic relationships between 47 proteins in these 7 species. A neighbor-joining tree was created by using MEGA X (1000 replicates), and the bootstrap value of each branch is displayed. At: *Arabidopsis thaliana*, Os: *Oryza sativa*, Dz: *Durio zibethinus*, Pt: *Populus trichocarpa*, Tc: *Therobroma cacao*, Cs: *Citrus sinensis*, Hb: *Hevea brasiliensis*. Black circles represent genes in *Arabidopsis*; black triangles represent genes in the rubber tree. Scale bar shows 0.1 amino acid substitutions per site.

**Figure 2 plants-12-01280-f002:**
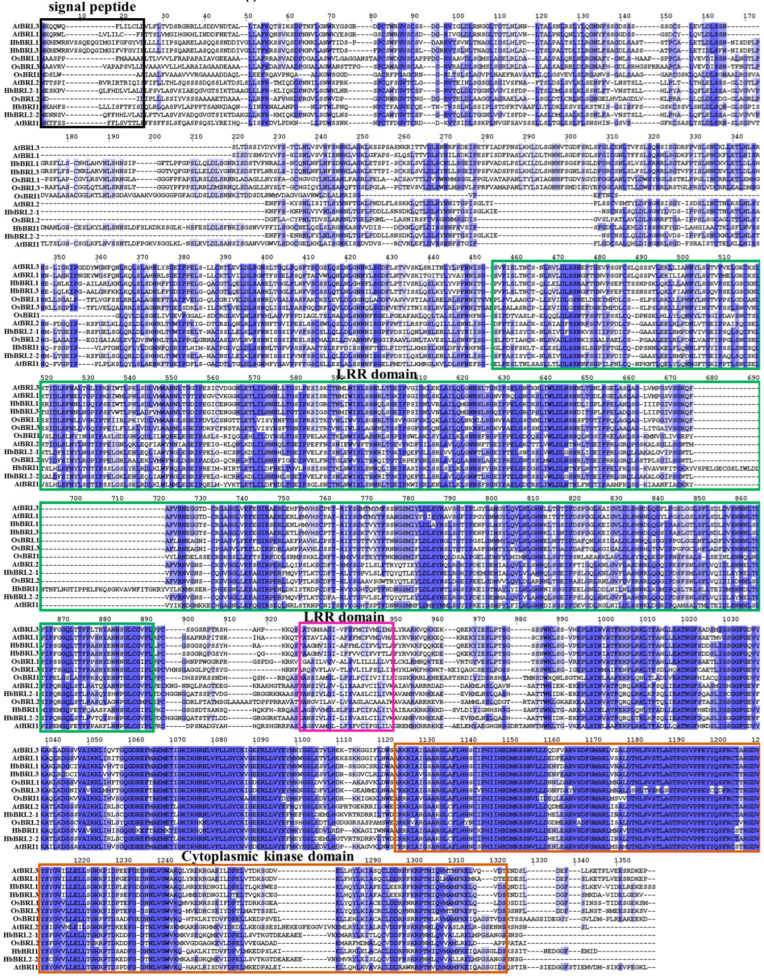
Multiple alignments of the HbBRI1 domains. Conserved domains of HbBRI1 amino acid sequences are marked in colored rectangles: signal peptide (black), LRR domain (green), transmembrane domain (rose red) and serine/threonine kinase domain (brown).

**Figure 3 plants-12-01280-f003:**
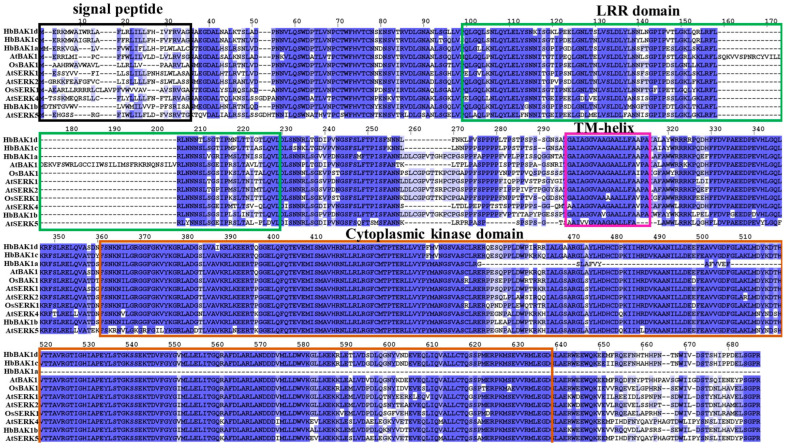
Multiple alignments of the HbBAK1 domains. Conserved domains of HbBAK1 amino acid sequences are marked in colored rectangles: signal peptide (black), LRR domain (green), transmembrane domain (rose red) and serine/threonine kinase domain (brown).

**Figure 4 plants-12-01280-f004:**
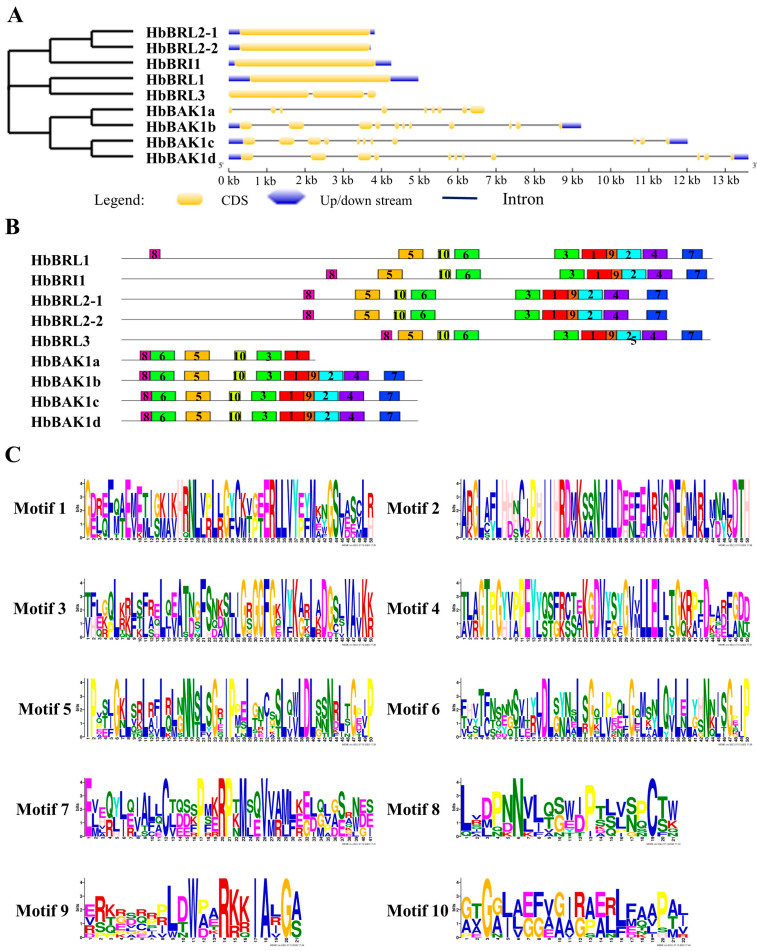
Gene structure and conserved motifs in five HbBRI1 and four HbBAK1 members. (**A**) Phylogenetic trees of HbBRI1 and HbBAK1 were compared by using MEGA X with the NL method. Gene structures of HbBRI1 and HbBAK1 genes were analyzed by GSDS. Yellow strips represent CDS, black lines delegate introns and blue strips indicate up/downstream. (**B**) Motifs in HbBRI1 and HbBAK1 members. Conserved motifs in HbBRI1 and HbBAK1, as executed by the MEME program, are displayed in the panel. Colored boxes represent different conserved motifs. (**C**) The sequence logo of the motifs. Motifs are shown by stacks of letters in each site. The height of the letters indicates that the letter appearing in the position is multiplied by the total amount of information on the stacks.

**Figure 5 plants-12-01280-f005:**
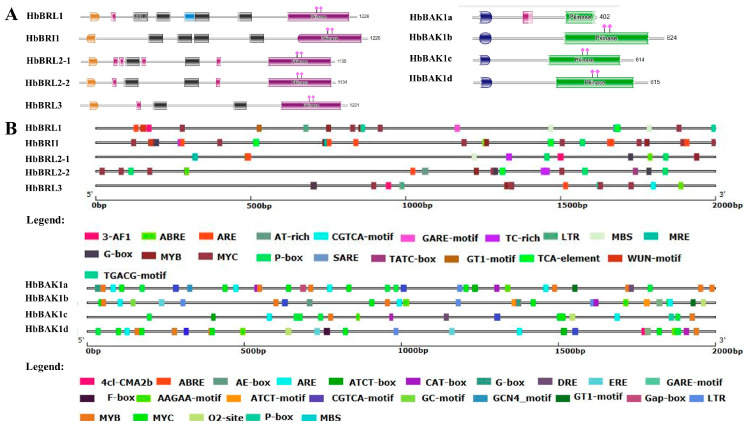
Conserved domain and cis-element analysis of BRI1 and BAK1 in the rubber tree. (**A**) Conserved domain of HbBRI1 and HbBAK1 proteins. All diagram regions before kinase domains are LRR domains; arrows pointing up in kinase domain denote the active site. (**B**) Cis-elements in *HbBRI1* and *HbBAK1* promoters. Up: cis-elements distributed in the promoter sequences of *HbBRI1s* and *HbBAK1s*. Down: different color boxes represent different element responses to different stresses.

**Figure 6 plants-12-01280-f006:**
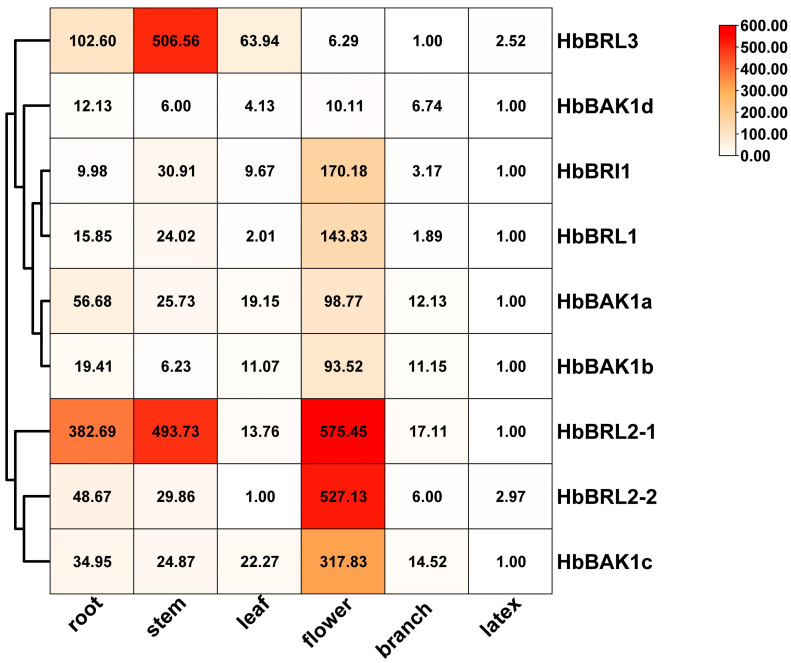
Expression patterns of HbBRI1 and HbBAK1 genes in different tissues. *HbActin* was the reference gene. The tissue with the lowest expression of each gene was the control, set as 1. Values in the box represent relative expression level. Expression values are shown in different colored boxes. White and red represent low and high expression levels, respectively.

**Figure 7 plants-12-01280-f007:**
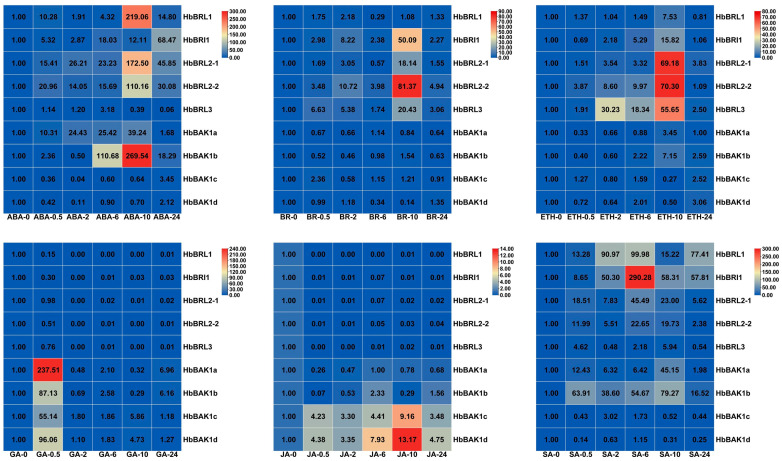
*HbBRI1* and *HbBAK1* response to different hormones (ABA, BR, ETH, GA, JA, SA). ABA, abscisic acid; BR, brassinosteroid; ETH, ethylene; GA, gibberellin; JA, jasmonic acid; SA, salicylic acid. Error bar indicates SD, which represents the standard deviation of three independent experiments. *HbActin* was the reference gene. Hormones treatment for temporal showed in X-axis. Expression values are shown in different colored boxes. Blue and red represent low and high expression levels, respectively.

**Table 1 plants-12-01280-t001:** Physical and chemical properties of BRI1 and BAK1 genes in the rubber tree.

Gene Name	Locus	Strand	CDS (bp)	Amino Acid (aa)	Molecular Weight	PI	GRAVY
*HbBRI1*	scaffold0387	+	3688	1228	134,151.21	6.19	−0.076
*HbBRL1*	scaffold0057	−	3679	1225	133,394.55	6.18	−0.041
*HbBRL2-1*	scaffold0406	−	3409	1135	124,018.58	5.94	−0.003
*HbBRL2-2*	scaffold0740	−	3406	1134	123,651.87	5.82	−0.002
*HbBRL3*	scaffold1656	−	3667	1221	132,827.64	5.48	−0.013
*HbBAK1a*	scaffold0043	+	1206	402	43,793.59	8.15	0.117
*HbBAK1b*	scaffold0283	−	1872	624	69,033.25	5.41	−0.135
*HbBAK1c*	scaffold0441	−	1845	615	68,225.60	5.76	−0.218
*HbBAK1d*	scaffold0577	−	1848	616	68,547.43	6.01	−0.199

## Data Availability

Data are contained within the article and Appendix A.

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
