# Peer review of "Comprehensive Analysis of BR Receptor Expression under Hormone Treatment in the Rubber Tree (*Hevea brasiliensis* Muell. Arg.)"

_plants, 2023, doi:10.3390/plants12061280_

Round 1

Reviewer 1 Report

In current manuscript submitted by Bingbing-Guo et al, the authors paid attention to BRI1 and BAK1, the receptors of Brassinosteroids (BR) in rubber tree. Based on bioinformatic prediction and analyses, five HbBRI1s and four HbBAK1s were identified in this study. The authors subsequently investigated the expression profiles of these genes in different tissues and under various hormonal treatments. In general, the subject is of great interests and importance to rubber tree biology. However, I feel the submitted data are preliminary and there are many issues need to be addressed. Please see my comments as below:

Major comments:

1. Issues on identification of HbBRI and HbBAK genes:

(1) The authors only select Arabidopsis and Rice genes to construct the phylogenetic tree as shown in Figure 1. However, I think more species (phylogenetically closer to rubber tree) and more genes if relevant should be included to increase the confidence level.

(2)       It would be problematic and misleading to name all rubber tree BRI-like genes as HbBRI1 since most of them are distant from AtBRI1 except for HbBTI1-2. Perhaps one could name others as BRI-like (BRL).

(3) In Figure 1, it would make more sense to label the gene names in addition to gene models for Arabidopsis and rice in the phylogenetic tree. What does LOC refer to for rice genes, please explain in the figure legend.

(4) In Figure 2, please include the amino acid sequences of BRI1 from Arabidopsis and rice for multiple alignment to confirm the conservation of major functional domains.

2. Expression analysis of HbBRI1 and HbBAK1 genes in different tissues were not properly performed.

(1) Since multiple tissues as well as multiple genes were investigated, I suggest an integrated heatmap for all data instead of several separated bar figures. Current format of data (Figure 5-7) and description in the text are not informative and difficult to follow. Moreover, “log2 fold change” could be better than the fold change given to the large variations.

(2) Since it is speculated that HbBRI1s and HbBAK1s played an important role in latex synthesis, one could expect to see a reasonable expression levels of these genes in related tissues. However, the reported tissue expression data seem not to support this, please elaborate this observation.

(3) The authors stated that “HbBRI1 and HbBAK1 genes displayed temporal and spatial specificity in different tissues”, however, no temporal data were present, please provide such data or revise the statement.

(4) Tissues used in qRT-PCR were not clearly reported. For instance, in Figure 6 and Supp. Table, the authors reported “latex” as one type of tissue. However, in the text, the authors mentioned “secondary laticifer” and “primary laticifer” as tissue types. Please clarify the exact tissues are used for qRT-PCR. Related to this, what does the “control” refer to when describing the fold change? Please point out what is the control tissue used in qRT-PCR quantification using the 2-△△Ct method?

(5) I cannot see the RPKM data from the rubber tree genome database nor the qRT-PCR primers.

(6) Materials and methods are lacking for tissue profile experiments. Are tissues from mature plants or seedlings used? How many biological replicates (individual plants) are used?

3. Issues in expression profile of HbBRI1s and HbBAK1s in response to hormones

(1) The authors didn’t report the expression levels of HbBRI1s and HbBAK1s in leaves in previous session. Please explain the relevance to use leaf materials for hormonal treatment assays.

(2) Since BRI1 and BAK1 are the receptors of BR, one could expect a primary response of these genes to BR treatment. First, are the expression patterns of identified HbBRI1s (Figure 6) and HbBAK1s (Figure 7) genes comparable with that in Arabidopsis or rice when BR treatment is applied? Second, please discuss the different patterns of HbBRI1s and HbBAK1s in response to BR.

(3) From Figure 6, seemed that HbBRI1-1 revealed a differential response pattern compared with other HbBRI1s. I wonder does this reflect the functional or structural divergency for HbBRI1-1.

(4) Again, I suggest a heatmap of gene expression profiles along time upon various hormonal treatments instead of separate bar figures in Figure 6 and 7.

(5) How did the authors normalize the data in Figure 6 and 7?

Minor comments:

(1) Please discuss the relations of HbBRI1 and HbBAK1 gene expression profiles (tissue specificity and hormonal response) to latex production in rubber tree.

(2) Please correct miss uses of abbreviations for genes and others throughout the manuscript. In many cases, gene names are not italicized.

(3) The author wrote in the text under the Results session that “In this study, total 5 and 2 complete overview of HbBRI1 and HbBAK1 genes were obtained in rubber tree using all kinds of bioinformatics resources…”, whereas in the abstract and table 1, 4 HbBAK1 were reported. Please confirm the number of HbBAK1 genes. In the same sentence, please explain, what does “all kinds of bioinformatics resources” mean?

(4) Please carefully check the languages, numerous linguistic mistakes are found in this manuscript. A professional proofreading is recommended.

(5) I suggest the authors make the discussion session more concise.

Reviewer 2 Report

This studies clonal analysis of HbBRI1 and HbBAK1 genes of rubber tree, as well as preliminary gene expression in Arabidopsis. The overall feeling is that something is missing and the figures presented is not clear; so the reviewer think the manuscript can be rejected or major revision.

1.       spaces are required; Line 53, 88, 218.

2.       There is gene cellular location in Table 1, but there is no corresponding figure(s) as cellular location support;

3.       Figure 2, the words in Figure 5 are too small to see clearly, can you enlarge the figure some?

4.       The words in Figure 6-7 are also too small to be clear at all.

5.       It shows the primary and secondary structure of the gene in the Figure 3, and whether the tertiary structure is predicted?

6.       Line 10, Rubber tree should add the botanical name;

7.       Line 107, Latin name should be italics.

Reviewer 3 Report

Review of manuscript entitled "Comprehensive analysis and expression under hormone of BR receptors in rubber tree (Hevea brasiliensis Muell. Arg.)".

In this manuscript authors characterize the BR-Insentive 1 (BRI1) and BRI1 Associated Kinase Receptor 1 (BAK1) in rubber tree, which are important kinases to transmit brassinosteroid signals to trigger various physiological functions.

Major points

It would be true to say that sequencing technology boost largely plant science as it can be seen from the increasing number of studies in the recent years. The genome/transcriptome/proteome of non-model organisms are practical and affordable for scientists but more essential issue would be why omics data of a given species has to be done and what is behind biological questions asked?

The characterization of BRI1 and BAK1 gene families in rubber tree and their domain/motif conservatives, cis-elements in promoter regions and tissue specificity and responding to hormones of each genes were reported in this study. However, I would like to ask authors to explain the contribution of these data set to what aspect of study using rubber tree?

Minor points

1. terminology: The word "cytomembrane" in line Line 38 might confuse the readers if you actually means "plasma membrane" for the location of both BRI1 and BAK1.

2. Font: The font type and size in table 1 were not identical. Please also consider the align the number to their decimal point in each column. In addition, please check carefully that Italics is required for scientific names and transcripts.

3. Typo: Please consider to check the manuscript word by word to avoid any typo in manuscript. For example: the "extrons" in Line 15 and the M"R"GA X in line 109.

4. Color code: Please consider to label different cis-elements with number instead of colors in figure 4B, the current version is hard to distinguish each of them, and please also rearrange 4B to increase the width of promoter and move the legend to the bottom of HbBRIs and HbBAK1. 

5. RT-qPCR: What is the internal control for authors to calculate 2-ΔΔCT as relative expression? It might be more clear to me if authors set the same scale of y-axis to compare the differences of HbBRIs and HbBAK1s among in different tissues. Please also group the same transcript responding to 6 hormones in to one bar chart and apply color codes to distinguish hormones among charts.

Round 2

Reviewer 1 Report

In the revised manuscript, the authors addressed most of the reviewer’s comments. I feel the manuscript has been improved to a great extent. However, I think there are a couple of aspects that need to be solved further.

-        Still, I think it will bring questions to name all HbBRI genes as BRI1. The authors took the published apple data as example to support the way they named the rubber genes, however, MdBRI1-5 (Zheng et al, 2017) not BRI1-1 to BRI1-5 were named in the published article. In addition, it was the authors’ own argue that “HbBRI1 (should be HbBRI1-1) was a homologous gene of AtBRL1 and AtBRL3 which were unnecessary combined to BRs”, suggesting that not all HbBRIs were closely homologous to AtBRI1 and functioning as a BR receptor. I suggest the authors re-consider this aspect.

-        Please double check the color code/value for Figure 5-6. Current color code is quite confusing. For example, in Figure 5, the highest value in the color code is limited to 10 (red) whereas in the heatmap, value around 10 is yellow. In Figure 6, the value of bar code is between -2 (blue) to 2 (red), however, no minus values were presented in the heatmaps.

-        In current Figure S1, the authors presented the expression data (RPKM value) for HbBRIs and HbBAKs from the genome database. However, it is unclear how the data were obtained, if the data were generated by the authors, the raw data should be submitted, otherwise, please provide the data source if published in elsewhere.

-        Line 308, I don’t see any data from the mentioned “microarray” in the manuscript, please explain.

-        Please correct the Latin name for Durian (Line 127)

-        Please check the format of reference, a couple of mistakes have been found, for example, ref. 69, 72 et al.

Reviewer 2 Report

Through the review, the author has revised the paper according to the requirements of the reviewer, and although the quality of the paper has not been significantly improved after the revision, at least the obvious errors of the paper have been reduced. Therefore, it is suggested that the manuscript can be accepted for publishment after minor revision.

The places that need minor repairs are:

The font on the right side in Figure 1 is too large, and it should be adjusted to be smaller one so that it is suitable for publication

Figure 5-6, the relative standard deviation on the right side of each small figure should be closer to the left side of the relevant figure.

Reviewer 3 Report

Dear Authors,

Many thanks for your careful revision and I do not have further request for current version of manuscript.

Author Response

Thank you for your approval of this manuscript.